# The Cysteine-Rich Peptide Snakin-2 Negatively Regulates Tubers Sprouting through Modulating Lignin Biosynthesis and H_2_O_2_ Accumulation in Potato

**DOI:** 10.3390/ijms22052287

**Published:** 2021-02-25

**Authors:** Mengsheng Deng, Jie Peng, Jie Zhang, Shuang Ran, Chengcheng Cai, Liping Yu, Su Ni, Xueli Huang, Liqin Li, Xiyao Wang

**Affiliations:** College of Agronomy, Sichuan Agriculture University, Chengdu 611130, China; dengmengsheng87@hotmail.com (M.D.); gusongke88@163.com (J.P.); ZhangjDMS@outlook.com (J.Z.); ranshuang98@163.com (S.R.); cccqyj@hotmail.com (C.C.); 757336519@163.com (L.Y.); ns13@163.com (S.N.); hxueli1983@163.com (X.H.)

**Keywords:** potato tuber, Snakin-2, dormancy, sprouting, lignin, hydrogen peroxide

## Abstract

Potato tuber dormancy is critical for the post-harvest quality. Snakin/Gibberellic Acid Stimulated in Arabidopsis (GASA) family genes are involved in the plants’ defense against pathogens and in growth and development, but the effect of Snakin-2 (SN2) on tuber dormancy and sprouting is largely unknown. In this study, a transgenic approach was applied to manipulate the expression level of *SN2* in tubers, and it demonstrated that *St*SN2 significantly controlled tuber sprouting, and silencing *StSN2* resulted in a release of dormancy and overexpressing tubers showed a longer dormant period than that of the control. Further analyses revealed that the decrease expression level accelerated skin cracking and water loss. Metabolite analyses revealed that *St*SN2 significantly down-regulated the accumulation of lignin precursors in the periderm, and the change of lignin content was documented, a finding which was consistent with the precursors’ level. Subsequently, proteomics found that cinnamyl alcohol dehydrogenase (CAD), caffeic acid O-methyltransferase (COMT) and peroxidase (Prx), the key proteins for lignin synthesis, were significantly up-regulated in silencing lines, and gene expression and enzyme activity analyses also supported this effect. Interestingly, we found that *St*SN2 physically interacts with three peroxidases catalyzing the oxidation and polymerization of lignin. In addition, SN2 altered the hydrogen peroxide (H_2_O_2_) content and the activities of superoxide dismutase (SOD) and catalase (CAT). These results suggest that *St*SN2 negatively regulates lignin biosynthesis and H_2_O_2_ accumulation, and ultimately inhibits the sprouting of potato tubers.

## 1. Introduction

Potato (*Solanum tuberosum* L.) is the third most important food and vegetable crop in the world, with wide adaptability, high yield and rich nutrition [1]. In 2018, the world’s total potato output exceeded 368 million tons, which meets the annual demand for planting, eating and processing. However, the potato industry is facing a serious problem regarding the inappropriate transition from dormancy to sprouting, which leads to the decline of tuber quality and marketability, and seriously affects the production, consumption and sales [1,2,3]. Tuber dormancy is established at the beginning of tuber formation and lasts for a relatively fixed period, and the length of the dormancy period will not be changed even under a suitable condition for sprouting [4]. Tuber dormancy is controlled by genotype and storage conditions pre- and post-harvest, but the specific mechanism of maintaining and releasing tuber dormancy is still unclear [2,4,5]. Therefore, to reveal the mechanism of tuber dormancy release is a very important issue in potato production, storage and processing.

Tuber dormancy release is a complex process that is known to involve several physiological and biochemical changes, mainly including hormonal signaling, carbohydrate metabolism and oxidation-reduction [4,6,7,8]. It is worth noting that, unlike rice and wheat seeds, tubers had a high moisture content (approximately 80%) and developed sprouts after a few months under normal conditions. Generally, cold storage at 2–4 °C is ideal for storing the seed potatoes but not for the potatoes for eating and processing. Worldwide, potatoes are being stored at 8–12 °C, and if treated with chlorpropham, camphor and/or essential oil at the same time, they will last up to 9 months [5,6,9]. However, the cost of cold storage is high, and CIPC has also been in control with a maximum residue limit of 10 mg·kg^−1^. In general, hormonal regulation and carbohydrate metabolism are thought to be the important factors controlling the dormancy release process in potato [4]. Very few studies have focused on the potential role of the periderm’s structure and components during this important physiological process. 

The complex periderm structure and its variable composition play a critical role in potato dormancy. The outermost shell consists of several strata of dead cells and forms the first natural line regulating dehydration and respiration. Noteworthily, the complex polyphenolic (lignin-like) and polyaliphatic in the phellem cell wall affected the tuber cortex and ultimately resulted in the change of dehydration and respiration [10]. In transgenic tubers with down-regulated expression of *FHT*, the skin appeared tawny and cracked, which contributed to the respiration with a 14-fold increase [11]. Moreover, the alteration in *FHT* level affected the content of ferulic acid, fatty acid and primary alcohol, and especially the ratio of guaiacyl/syringyl in tuber wound healing [10]. Similarly, the knockdown of *SlDCR* resulted in the disrupted epidermal layer of tomato with changes of lignin, suberin, phenylpropanoids and fatty acids [12,13]. Many investigations indicated that polyphenolic exhibited important roles in regulating fruit skin cracking and ensuring superior quality and freshness [12,14]. Vulavala et al. revealed that lignin and suberin are involved in potato periderm formation and maturation, and that the phellogen may affect the incomplete skin-set and skin injuries [15]. Additionally, in Arabidopsis, the decrease of lignin in the testa inhibited seed germination [16,17]. In short, lignin may regulate tubers’ respiration, freshness and dormancy in potato.

A previous study has found that StSN2, a member of the Snakin/GASA family, is positively correlated with tuber dormancy, and that *StSN2* expression level decreased more slowly in the long-dormant cultivars, suggesting that StSN2 might maintain tuber dormancy [6]. Snakin/GASA members are widely known to be involved in seed dormancy, leaf morphology, flowering and biotic and abiotic stresses [18,19,20,21,22,23,24,25]. Nahirñak et al. found that silencing *SN1* shortened plant height and altered the leaf shape in potato, which was accompanied by an alteration of cell wall composition and primary metabolism associated with cell division, whereas a non-significant difference in overexpressing lines was found, affecting reactive oxygen (ROS) and hormone balance including gibberellic acid (GA), brassinosteriod (BR) and salicylic acid (SA) [20,24]. It was reported that *AtGASA6* served as an integrator of gibberellic acid and abscisic acid (ABA), and stimulated germination through *EXPA1* by promoting cell elongation and increasing the hypocotyls’ length [22]. Similarly, *AtGASA4* promoted GA-induced germination, but *AtGASA5* showed opposite effects [19,26]. Therefore, Snakin/GASA exhibited flexible functions in growth and development. 

Potato dormancy is regulated by multiple genes, which are mainly related to hormone and sugar metabolism, while the effects of other aspects on tuber dormancy are more interesting and need to be further explored. The present study reported the function of plant antimicrobial peptide StSN2 in maintaining tuber dormancy. Transgenic tubers with a dormant period of a different length were obtained by changing the expression level of *StSN2*. We analyzed the glossiness and integrity of the tuber surface and the morphology and arrangement of periderm cells. A comprehensive analysis suggested that the significant physiological changes occurred in tubers at ~45 days, and at this stage, tubers were subjected to proteomic and metabolomic analysis. The results showed that silencing SN2 promoted the accumulation of lignin precursors and induced lignin-related protein expression in the tuber periderm. The lignin synthase, cinnamyl alcohol dehydrogenase (CAD), caffeic acid O-methyltransferase (COMT1), peroxidase (Prx) and Cinnamoyl-CoA reductase (CCR) significantly increased in the silencing periderm at the transcriptional level and enzyme activity, which finally led to the accumulation of lignin [27,28,29,30,31]. The crack and respiration resulted from much lignin and stimulated the release of tuber dormancy. This research will contribute to a better understanding of the effect of StSN2 on lignin synthesis and tuber dormancy.

## 2. Results 

### 2.1. StSN2 Expression Level and Phenotypic Characterization of StSN2 Transgenic Lines

In our previous studies, transcriptome and proteome analyses indicated that the level of SN2 is closely related to the maintenance of tuber dormancy. Further, qRT-PCR analysis confirmed that the expression of SN2 is positively related to tuber dormancy. Therefore, to explore the function of SN2 in regulating tuber dormancy, different *SN2* constructs were developed to change the expression of *SN2* in tubers, and 23 RNAi lines and 27 over-expression lines (OE) were obtained. RNAi lines 7 and 8, and OE lines 11 and 27 were randomly selected for the following experiments. 

Firstly, the dormancy characteristic of different transgenic potatoes were analyzed by comparing them with the control (wild-type and empty vector transformants with same traits). No sprout was observed in any tuber within 30 d. However, there were significant differences in the dormancy period of the three groups (OE, ~105 d, WT, ~90 d, RNAi, ~75 d). During 30–75 d, the two over-expression lines showed slow sprouting: the highest sprouting rate of overexpressing line 27 was only 41.55% at 75 d, while 87.44% of RNAi line 7 have sprouted at the same time (Figure 1A). Photographs of the sprouting in the potatoes were taken after 15 and 75 d (Figure 1B). 105 days later, the sprouting rates of two over-expression tubers with a long dormancy period were still the lowest compared with *RNAi* tubers and wild-type, and were 79.70% and 82.90%, respectively. Conversely, the RNAi lines and wild-type tubers sprouted completely at 105 d (Figure 1A). The results suggested that overexpressing SN2 could significantly inhibit tuber sprouting.

The SN2 expression level was detected in an apical sprout by using qRT-PCR. In over-expressed tubers, the level was significantly increased, and became 3.63–4.71 times higher than that of the control. Moreover, the high level was completely consistent with their long dormancy period. For example, overexpressing tubers with high *StSN2* level showed that only 95.21% of the tubes sprouted as late as 130 d. To analyze the expression strength of *SN2* in RNAi lines, we measured the *SN2* expression level of lines 7 and 8 as 30.08% and 20.79%, respectively, in comparison with the control (Figure 2A). In addition, a similar tendency was obtained at the protein level (Figure 2B). 

### 2.2. Effect of StSN2 on Cracking and Cell Morphology of Periderm and Water Loss

During 120 d storage, the weight of the tubers in each line decreased gradually, but the reduction rates were significantly different. The weight of the overexpressing tubers always remained at the highest level comparing with other groups, amounting to 52.4 g and 55.6 g at 120 d, respectively. However, the silencing *StSN2* tubers, in RNAi line 7 and RNAi line 8, the suffered a rapid weight loss at 0–60 d, and the loss rate slowed down after sprouting (60–120 d). In the end, the weights were only 19.7 g and 23.3 g at 120 d respectively (Figure 3A). 

Compared with the overexpression and wild-type, the silencing tubers’ skin was mostly cracked, rough and brownish, and was photographed at 30 days (Figure 3B,C). Microscopy analyses of the tuber skin cells revealed that the cells of RNAi line 7 were deformed and wrapped in the necrotic cellular debris, especially at the junction with cells, which was possibly caused by the excessive water loss of the tubers (Figure 3D). Consistently, at the 45th day of storage, a significant decrease in the thick of the phellem cell layers was observed in the silencing line, and the outer 3–4 cell layers had completely collapsed. Whereas, like the fresh tubers, the phellem cells in overexpressing tubers were more orderly and turgid due to having enough water (Figure 3E). 

### 2.3. Skin Metabolome in StSN2 Transgenic Potato

Potato tubers as storage organs necessitate mechanisms to reduce water loss and regulate transpiration and respiration, and secondary metabolites such as lignin form an important barrier for these physiological processes. Different skins from RNAi, wild-type and overexpressing tubers were subjected to LC-MS, and 371 differentially accumulated metabolites (DAMs) were identified, including amino acids, organic acids, phenols and other secondary metabolites (Appendix A). Further analysis revealed that, among the DAMs, 30 lignin-related metabolites were screened and mainly classified into “Phenylpropanoid biosynthesis”, “Phenylalanine, tyrosine and tryptophan biosynthesis” and “Phenylalanine metabolism” by Kyoto Encyclopedia of Genes and Genomes (KEGG) analysis. Based on the change intensity, a hierarchical clustering analysis divided the candidate metabolites into four groups (Figure 4A). The contents of nine DAMs (Class I) were significantly higher in the silencing line than in the overexpressing line and WT, while the contents of two DAMs (Class IV) were lowest in the silencing line. To our surprise, the majority of DAMs (Class II and Class III) shared the same tendency in the transgenic lines (Figure 4A). Nevertheless, the key metabolites for phenylpropanoid biosynthesis, such as 4-hydroxycinnamic acid, ferulic acid and the monolignols coniferyl alcohol and sinapyl alcohol, were the most abundant in the RNAi line, providing the sources of lignin synthesis and accumulation (Figure 4B–E). Furthermore, silencing *StSN2* resulted in a 40.45% and a 15.51% increase in lignin content in two transgenic lines, respectively, while in the overexpressed phellem the lignin content was only 24.27–30.94% of that in the wild-type (Figure 4F).

### 2.4. Proteomics Analysis of Lignin Biosynthesis 

To better understand the molecular mechanisms of StSN2 regulating lignin synthesis, a protein expression profile was obtained from transgenic periderm by LC-MS/MS analysis at the 45^th^ day of storage (Appendix A). 382 differentially expressed proteins (DEPs) were identified based on the thresholds (fold change >1.2 or <0.83). The KEGG pathway annotation analysis of DEPs from each comparison showed that they were enriched in a number of metabolic pathways, including carbohydrate metabolism, amino acid metabolism and other secondary metabolism (Appendix A). Subsequently, we focused on the pathway enrichment analysis of the DEPs in the three comparison groups (RNAi7 vs. WT, RNAi7 *vs.* OE27 and OE27 vs. WT), and the results indicated that 15 DEPs between RNAi line 7 and WT were enriched in “Phenylpropanoid biosynthesis”, “Phenylalanine, tyrosine and tryptophan biosynthesis” and “Phenylalanine metabolism”, and 11 DEPs from RNAi line 7 vs. OE line *27* and 4 DEPs from OE line 27 vs. WT were enriched in “Phenylpropanoid biosynthesis” (Figure 5A–C). Moreover, the critical pathway for lignin synthesis “Phenylpropanoid biosynthesis”, including peroxidases, CAD and COMT, was up-regulated in RNAi line 7 by comparison with overexpressed and wild-type tubers (Appendix A). It is worth noting that a total of nine peroxidases were identified, and the levels of five proteins (Prx1, Prx2, Prx4, Prx5, Prx6) were positively regulated by StSN2, while those peroxidases, from seven to ten, were highly expressing in silencing periderm. The rest of the proteins altered irregularly (Appendix A).

To determine whether the StSN2-altered expression affected the transcription and enzyme activities of lignin biosynthesis key genes, the following steps were taken. Firstly, the qRT-PCR analysis indicated that the expression level of *COMT* and *CAD* was increased by ~5 to 10-fold in the periderm of silencing lines compared with that of the wild-type, but there was a slight change in overexpressing lines. Strikingly, the expression of *Prx10* was significantly different among all lines, and the highest one was RNAi line 7 (Figure 5D). Finally, we focused on the changes in enzyme activities and found that the enzyme activities of peroxidases, COMT and CAD in silencing tubers were significantly higher than that of the wild-type, but decreased in overexpression tubers except for CAD (Figure 5E). Similarly, we found the inhibitory effect of StSN2 on another lignin biosynthesis enzyme CCR. In summary, these results suggested that StSN2 inhibited lignin biosynthesis by mainly regulating the expression and activities of several key enzymes. 

### 2.5. StSN2 Interacts with Three Class III Peroxidases

Preliminary studies indicated that GASAs were involved in regulating plant growth and development mainly through interacting with proteins. For example, StSN1 was found to interact with DIM/DWF1 and SUT1 [24,32]. Therefore, co-immunoprecipitation (Co-IP) and mass spectrum (MS) were applied to explore the proteins interacting with StSN2 and regulating lignin synthesis in RNAi line 7 and OE line 27, and the results showed that three class III peroxidases, including Prx2, Prx 9 and Prx 10, can interact with StSN2 in vitro, which catalyzed the oxidation and polymerization of lignin (Appendix A, Appendix A). Furthermore, a yeast two-hybrid assay confirmed that StSN2 interacted with Prx2, Prx 9 and Prx 10 (Figure 6). Interestingly, the same expression tendency of these three proteins was detected in proteome and CoIP-MS data, and the high level of these peroxidases may contribute to lignin synthesis (Appendix A). The results indicated that StSN2 physically interacted with peroxidases to regulate lignin synthesis.

### 2.6. StSN2 Altered H_2_O_2_ Content and the Activities of Superoxide Dismutase (SOD) and Catalase (CAT)

Hydrogen peroxide is an important condition for lignin synthesis and promotes potato tuber sprouting [33,34,35]. The level of H_2_O_2_ in the bud was analyzed, and the results showed that StSN2 significantly inhibited the accumulation of H_2_O_2_, except for OE line 11 (Figure 7A). Moreover, this difference was also supported by the results of SOD and CAT activity analysis, and especially the decrease of CAT activity in RNAi line 7, which may play an important role in maintaining a high level of H_2_O_2_ in the bud (Figure 7B,C).

## 3. Discussion

In the potato life cycle, tuber dormancy and its release are the key physiological processes and are under genetic and environmental control [8]. However, unlike seeds such as rice and corn, potato with high water content sprout easily under right conditions [1]. Perhaps the storage of potato is more similar to that of fresh fruits with buds, so we need to consider both controlling sprouting and keeping them fresh. Previously, we screened the *SN2* gene, which was positively related to dormancy through proteome and transcriptome analysis, and a qRT-PCR analysis also showed that *SN2* was highly expressed in deep dormancy tubers [6]. Previous reports indicated that Snakin/GASA played an important role in regulating seed dormancy and germination, and were involved in cell division, hormone balance, ROS and other processes [20,22,24,26,36,37,38,39]. It was also found, in potato, that StSN1 played a role in cell elongation and hormonal signaling, although there is no direct evidence to support SN2 maintaining dormancy [20,24]. Surprisingly, the expression level of *SN2* in tubers was found to cause a significant difference in the length of the dormancy period of tubers (Figure 1 and Appendix A). In addition, qRT-PCR and western blot analysis suggested that SN2 is mainly expressed in tubers rather than in the stem and the leaf (Appendix A), and SN2 also played an important role in the process of tuber wound healing (Figure 4A and Appendix A). The results showed that SN2 had important biological functions in regulating dormancy in tubers.

Most Snakin/GASA genes have been reported to be induced by hormones. In Arabidopsis, *GASA4* and *GASA6* were induced in general by GA and BR and repressed by ABA, and GASA6 participated in the antagonistic regulation of seed germination by GA and ABA, and stimulated EXP1 expressing in cell walls [22,23]. Sun et al. proposed that *GASA14* was essential for GA-induced germination and inhibited by GAI and RGL [21]. However, other members of Snakin/GASA exhibited different modes and functions. The previous study suggested that *StSN2* is induced by ABA but inhibited by GA [40]. Similarly, in potato, the same tendency was found in the *SN2* expression level (Appendix A), and it demonstrated that SN2 was locally up-regulated in ABA-induced wounding signaling [41]. Further, in potato dormancy, the level of *SN2* was significantly induced by ABA, but repressed by GA and BR, indicating that SN2 may also be involved in the maintenance of tuber dormancy mediated by hormones (Appendix A). 

On the other hand, Snakin/GASA participated in the cross-talk of hormones including ABA, GA, BR and SA. OsGSR1 was found to be involved in the crosstalk of GA and BR and to interact with DIM/DWF1 in rice [42]. And in potato, SN1 affected hormone balance by interacting with DIM/DWF1 and decreasing the SA and GA content [24]. Moreover, under the treatment with GA and BR, the expressions of SN2 were all down-regulated, indicating that SN2 may negatively participate in the synergism of GA and BR in sprouting (Appendix A). However, the interaction between Snakin/GASA and hormonal proteins was found in BR signaling rather than in GA and ABA signaling. Subsequently, the yeast two-hybrid assay was applied and demonstrated that SN2 did not interact with DIM/DWF1. Conversely, we identified the physical interaction between SN2 and ABA signaling factor SnRK2.4 by CoIP-MS analysis (Appendix A, Appendix A), which was combined with the finding that overexpressing SN2 enhanced the level of SnRK2.4 and the same variation in *ABI3* and *ABI5* (Appendix A, Appendix A). The results showed that SN2 played a positive role in the maintenance of tuber dormancy by ABA. 

The Snakin/GASA family, including GIP2, GASA5 and GASA14, has a conservative domain with 12 cysteine amino acids, which are involved in redox homeostasis [40]. In potato, Nahirñak et al. detected that SN1-silenced inhibited ROS accumulation in leaves and reduced the content of ascorbic acid [24]. Similarly, SN2 included highly conserved amino acid sequences (CX3CX3CX8CX3CX2CCX2CXC X11CXCX12CX) (Appendix A) and down-regulated the level of H_2_O_2_ with the change of Prx, CAT and SOD activities in the bud (Figure 7). It is well known that exogenous H_2_O_2_ application induces the breaking of dormancy, which has been identified both in plant seeds and in vegetative buds [34,35]. Liu et al. reported that potato dormancy break needs rely on ROS, and the NADPH oxidase inhibitor (DPI) significantly affected the sprouting rate and down-regulated the level of ROS [4].Consistently, the qPCR assay revealed that *RobhA* and *RobhB*, the key genes that catalyzed the production of ROS, were up-regulated in the silencing bud. As mentioned above, SN2 probably has an important role in potato dormancy and its release. In our study, a SN2-regulated high level of H_2_O_2_ stimulated the total peroxidases activities (EC 1.11.1.7), but, in fact, not all of the eight differential peroxidases increased in silencing buds (Figure 7). In addition, peroxidases can not only catalyze the production of ROS, but also of scavenging reactive oxygen species, so this contradiction mechanism is still unclear, and peroxidases have functional redundancy in regulating plant physiology. However, the important point we focus on is that peroxidases catalyzed the oxidative polymerization of lignin, and H_2_O_2_ as oxidant is essential for it.

The periderm is the natural barrier for potato development, and its composition and structure are very important for transpiration and respiration during potato dormancy [43]. A set of transgenic tubers with rough skin was developed and exhibited decreased levels of SN2 expression (Figure 1 and Figure 2). Preliminary results demonstrated a clear effect of the manipulation of SN2 expression on the skin and periderm. It has been confirmed that Snakin/GASA is involved in cell division, elongation and cell wall formation. In potato, silencing *StSN1* inhibited cell division and growth in leaves by changing the cell wall composition [24]. On the contrary, overexpressing *AtGASA6* enhanced the expression of expansins, promoted hypocotyls cell elongation and stimulated seed germination [22]. In general, cell wall-related Snakin/GASA are located on the cell wall, such as AtGASA10 and FaGAST1, while SN1 is located on the plasma membranes rather than the cell wall [20,44,45]. Although we have not analyzed the location of StSN2, we have found that the effect of StSN2 on periderm cells is manifested in the cell’s arrangement and shape, which seemed to have a weak effect on cell division and elongation, but we still observed the existence of cells with different sizes (Figure 3).

In previous studies, the skin of fruits became rough and cracked, which was related to aromatic compounds, phenylpropanoids, lignin and fatty acids. As in the case of *RNAi-DCR* tomato and “Rugiada” apple fruit, the surface of *RNAi-StSN2* potatoes appeared brown, rough and cracked [12]. We subsequently performed a characterization of the cracked *RNAi-StSN2* potato (Figure 3). The results showed a significant increment in the majority of the quantified lignin monomers, notably a massive increase in the contents of coniferyl alcohol and sinapyl alcohol (Figure 4). In follow-up experiments, the single most striking observation to emerge from the data comparison was that the expression and activity of key enzymes of lignin synthesis, CAD, Prx and COMT, increased with the down-regulation of *StSN2* expression level, and the effect of *StSN2* silencing on them was more significant than that of the overexpressing lines (Figure 5). Similarly, the alteration of *StSN2* level resulted in significant changes in the content of lignin and its precursors, and association analysis revealed that two highly correlated groups were “COMT-caffeic acid, ferulic acid” and “Prx-Coniferyl Alcohol, Sinapyl Alcohol”, respectively (Appendix A). Furthermore, in *Arabidopsis*, Liang et al. mutated laccase 15 to change the content of lignin in coat, and found that the low level of lignin contributed to seed germination [16]. It was also found that the germination rate of *prx2prx25* mutant seeds is higher than that of wild-type seeds in *Arabidopsis*, and mainly because peroxidase affects the polymerization and the level of lignin [17]. Meanwhile, in Citrus sinensis, the overexpression of CsPrx25 enhanced H_2_O_2_ levels and cell wall lignification [46]. In SN2-RNAi potato, the high lignin level resulted in periderm collapse and was more conducive to tuber sprouting, and CoIP-MS and yeast two-hybrid analysis showed a strong interaction between peroxidases and StSN2, indicating that StSN2 regulated lignin synthesis by interacting with peroxidases (Figure 6, Appendix A). Therefore, one important aspect of lignin mediated by StSN2 is that it plays a vital role in potato cracking and preservation.

In conclusion, we identified a pivotal gene, StSN2, related to dormancy, which maintains tuber dormancy mainly in two patterns, as follows. Firstly, StSN2 inhibited the accumulation of hydrogen peroxide in bud tissue and negatively regulated its effect of stimulating tuber sprouting. Secondly, on the basis of inhibiting the accumulation of hydrogen peroxide, StSN2 further affected the biosynthesis of lignin and tuber skin, and regulated the water loss and respiration, finally achieved in the control of sprouting and preservation. The empirical findings in this study provide a novel insight into the mechanism whereby StSN2 maintains dormancy. 

## 4. Materials and Methods

### 4.1. Plant Material and Growth Conditions

The aseptic plantlets were propagated in 60 mm glass bottles containing Murashige and Skoog with 15 g/L sucrose and 7 g/L agar at 20 ± 1 °C, 16 h light, 8 h dark, light intensity 100 µmol m^−2^ s^−1^. Transgenic plantlets were selected for the resistance to 50 mg/mL kanamycin. Two weeks later, the robust tissue culture plantlets, with a height of ~5 cm, were transplanted into 18 cm pots containing peat soil, under the condition of natural light and 16 h light (21 °C) and 8 h dark (18 °C). The leaves were sprayed with Hogland nutrient solution at 3 and 6 weeks. When seedlings turned yellow, at about 10 weeks, the harvested tubers with the same maturity were kept in darkness at 13–15 °C for 10-day wound healing, then used for morphological observation and subsequent index determination.

### 4.2. Generation of StSN2 Transgenic Potato Lines 

The dormancy gene *SN2* (Soltu.DM.01G050660.1), a member of the Snakin/GASA family, was screened from transcriptome and proteome data [6]. To generate *StSN2* constructs for overexpressing transgenic potato lines, primers were designed to amplify a 315 bp ORF (open read frame) from tuber cDNA prepared from potato cultivar *Chuanyu 10*. *Xba*I and *Sma*I sites were engineered at the start and the termini, respectively. The recombinant vector pBI121-SN2 was constructed under T4 ligase, then digested with restriction enzymes as described above to check the accuracy. Similarly, a 315 bp StSN2-specific fragment was amplified by PCR and used in a restriction-ligation reaction for insertion into the binary vector pBI121. Details of the primer sequences used for cloning are provided in Appendix A. The recombinant expression vectors were transformed into Agrobacterium strain GV3101 by the frozen-thawed method. The stem segments of the 7-day in vitro culture were infected and transformed as described previously [47]. Finally, the aseptic plantlets with different levels of *StSN2* were obtained.

### 4.3. Western Blot and Quantitative Real-Time PCR

Before Western blot, the antibodies from rabbit serum were purified using the respective oligopeptides as the affinity column tag [8]. Western blot was performed to detect protein levels in potato bud eyes. Approximately 10 μg protein was loaded per lane, then 5% non-fat milk was used to block non-specific protein binding, and the nitrocellulose membrane was incubated with StSN2. For the quantitative real-time PCR (qPCR) analysis, the total tuber RNA was isolated using TRIzol reagent (Invitrogen, Carlsbad, USA) according to the manufacturer’s protocol. qPCR was then performed on a 7500 Real Time PCR System (Life Technologies) according to the manufacturer’s instructions. The 2^−ΔΔCt^ method was used for relative quantification. *Elongation factor 1 alpha-like* (*EF1αL*) expression was used as an internal control. Three biological replicates and three technical replicates were performed for all experiments.

### 4.4. Assessment of Sprout Growth, Periderm Morphology and Tuber Weight 

Tubers from multiple transgenic lines were assessed for tuber sprout growth. Once harvested, the healthy tubers were placed under scattered light at room temperature for wound healing and then transferred to a dark and relatively closed carton and stored at 15 ± 2 °C and a relative humidity of 65% ± 5%. The sprouting rate and weight of each transgenic line were measured every 15 d, and photos were taken. The changes in the tuber periderm morphology (such as color and glossiness) were observed periodically under the prompt microscope, and the thickness, cell arrangement and cell size of different transgenic tubers were monitored by light microscopy. The tubers fixed above 24 h were dehydrated with alcohol with different concentrations. After being embedded, periderms about 4 μm thick were observed under the light microscope. For the scanning electron microscopy (SEM), small fragments of tuber periderm were fixed with 1% osmic acid in PBS (pH 7.4) at room temperature for 1–2 h. Fragments were dehydrated with an increasing ethanol concentration series, exchanged through isoamyl acetate and critical point-dried. The fragments were attached to the conductive carbon double-sided adhesive and coated with gold for 30 s. Specimens were observed using the Hitachi SU8100 SEM.

### 4.5. Targeted Metabolomic Analysis on Metabolites in Potato Skin

Tissues (100 mg) were resuspended with 80% methanol and 0.1% formic acid by well vortexing. Some of the supernatant was diluted to a final concentration containing 53 % methanol by LC-MS grade water. After centrifuged, the supernatant was injected into the LC-MS/MS system analysis. LC-MS/MS analyses were performed using an ExionLC™AD system (SCIEX, Shanghai, China) coupled with a QTRAP® 6500+ mass spectrometer (SCIEX, Shanghai, China). The samples were injected onto aHSS T3 Column (100 mm × 2.1 mm) using a 25-min linear gradient at a flow rate of 0.35 mL/min for the negative polarity mode. Next, metabolites were identified and quantified using MRM (Multiple Reaction Monitoring).

### 4.6. Identification and Quantification of Proteins

The total proteins were extracted from WT, RNAi line 7 and OE line 27. The concentration of the total protein was determined by a Bradford protein quantitative kit, and the quality of the proteins was assessed by SDS-PAGE. Then, 3 μL 1 μg/μL trypsin (Promega, Madison, WI, USA) and a 500 μL 50 mM TEAB buffer (Triethyl ammonium bicarbonate) were used to digest the proteins of each sample overnight at 37 °C, and the products purified by the C18 desalination column were labeled with TMT. Next, we used an L-3000 HPLC system to gradient elute the solution containing protein powder, and 10 fractions were obtained. The chromatographic column was Waters BEH C18 (4.6 × 250 mm, 5 μm), and the column temperature was set at 50 °C. For transition library construction, shotgun proteomics analyses were performed using an EASY-nLCTM 1200 UHPLC system (Thermo Fisher, Waltham, MA, USA) coupled with an Q Exactive HF-X mass spectrometer (Thermo Fisher) operating in the data-dependent acquisition (DDA) mode. A 1-μg sample was injected into a home-made C18 Nano-Trap column (2 cm × 75 μm, 3 μm). Peptides were separated in a home-made analytical column (15 cm × 150 μm, 1.9 μm), using a linear gradient elution. The separated peptides were analyzed by a Q Exactive HF-X mass spectrometer (Thermo Fisher). The resulting MS/MS data were processed using Proteome Discoverer 2.2 (PD 2.2, Thermo).

### 4.7. CoIP-MS Assays

An anti-StSN2 antibody was prepared by polypeptides (SIQTDQVTSNAISEA) in a rabbit, and used to co-immunoprecipitate SN2-interacting proteins [48]. For in-gel tryptic digestion, gel pieces were digested with trypsin at 37 °C overnight after destained and dehydrated [49]. Then, peptides were extracted with 50% acetonitrile/5% formic acid, followed by 100% acetonitrile. The peptides were dried to completion and resuspended in 2% acetonitrile/0.1% formic acid, whereafter these unknown peptides were subjected to LC-MS/MS analysis to identify candidate proteins’ name and function.

### 4.8. Yeast Two-Hybrid Assay

The full-length of *SN2* was fused to the bait vector pGBKT7, and the AD vector (pGADT7) was used to express peroxidases. Then, both pairs of plasmids (SN2-BD/Prxs-AD) were co-transformed into the yeast strain AH109 (Coolaber). The co-transformation colonies were selected on SD-Trp-Leu plates. Positive clones were transferred and grown on SD/-Leu-Trp-His plates, and the β-Galactosidase activity was measured. SN2-BD and pGADT7 were used as negative controls. AtTOPP4 and AtPIN1 were used as positive controls [50]. The primers used for generating various clones in this study are listed in Appendix A.

### 4.9. Measurement of Lignin Content and Enzyme Activities

The extraction and quantification of lignin and H_2_O_2_ were performed based on the procedure described in the manufacturer’s directions (Solarbio, China). The activities of Prx, SOD, CAT and CAD were measured separately by using a PRX assay kit (Cat. BC0095), SOD assay kit (Cat. BC0175), CAT assay kit (Cat. BC0205) and CAD assay kit (Cat. BC4170) produced by Solarbio life science. All samples were prepared for enzyme activity by homogenizing 0.1 g of tuber in a solution of 0.01 mM PH 7.2 phosphate buffer saline. The homogenate was centrifuged at 12 000 rpm for 10 min at 4 °C. The activity of CCR and COMT were determined by the assay kit of Bioroyee (Beijing, China).

### 4.10. Statistical Analysis

For all generated data, at least three biological replicates were performed for each sample. The data were subjected to unpaired Student’s t-tests with *p* ≤ 0.01 and *p* ≤ 0.05. Data are shown as the mean ± SE (*n* = 3), and n represents the biological replicates. Excel 2019 (Microsoft Corporation, Redmond, WA, USA) and the SPSS 14.0 software (IBM, New York, NY, USA) were used for statistical analysis. The statistical results were reported as the mean ± SD. 

## Figures and Tables

**Figure 1 ijms-22-02287-f001:**
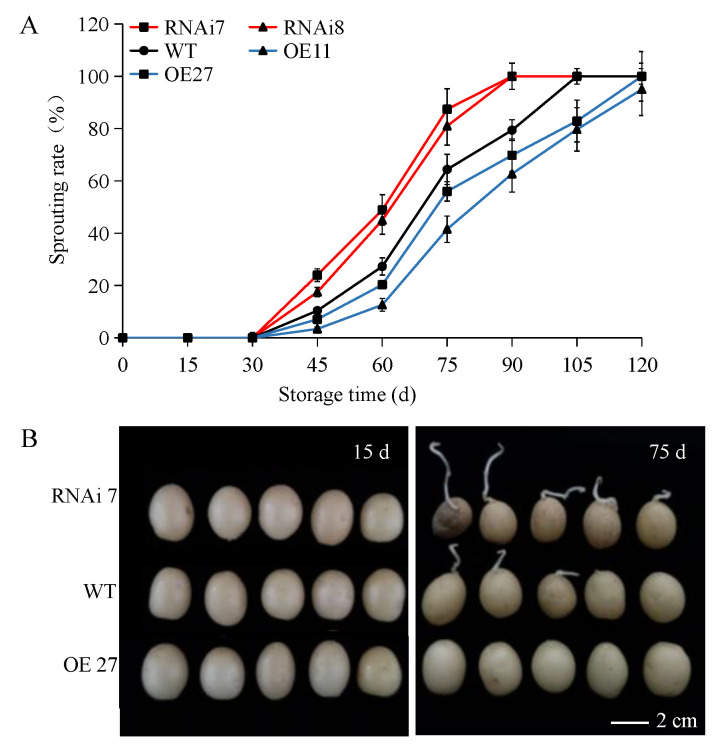
Sprouting rate and phenotypes of *StSN2* transgenic tubers and controls during different storage times. (**A**) Statistical results of sprouting rate. (**B**) Sprout phenotypes on the 15th and 75th day of storage. Data are means ± SD of three biological replicates. Scale bar = 1 cm.

**Figure 2 ijms-22-02287-f002:**
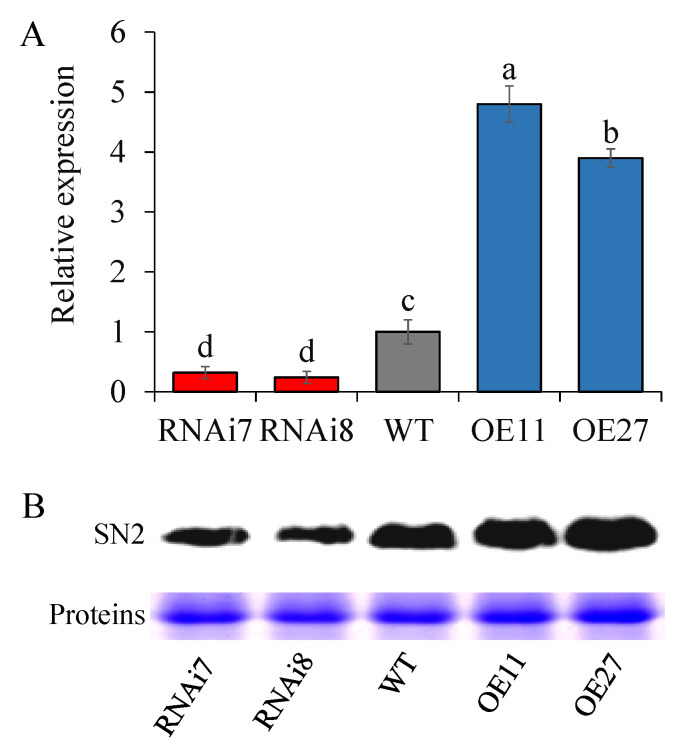
Comparison of gene and protein expression levels in sprouts of wild-type (WT) with transgenic plants either down-regulating (RNAi) or overexpressing (OE) the *StSN2* gene. (**A**) Gene expression levels in transgenic lines. Data are means ± SD of three biological replicates. Different letters indicate significant differences at *p* < 0.05. (**B**) Protein expression levels measured by western blot (WB). The wild-type (WT), potato cultivar *Chuanyu 10*, was used as the control.

**Figure 3 ijms-22-02287-f003:**
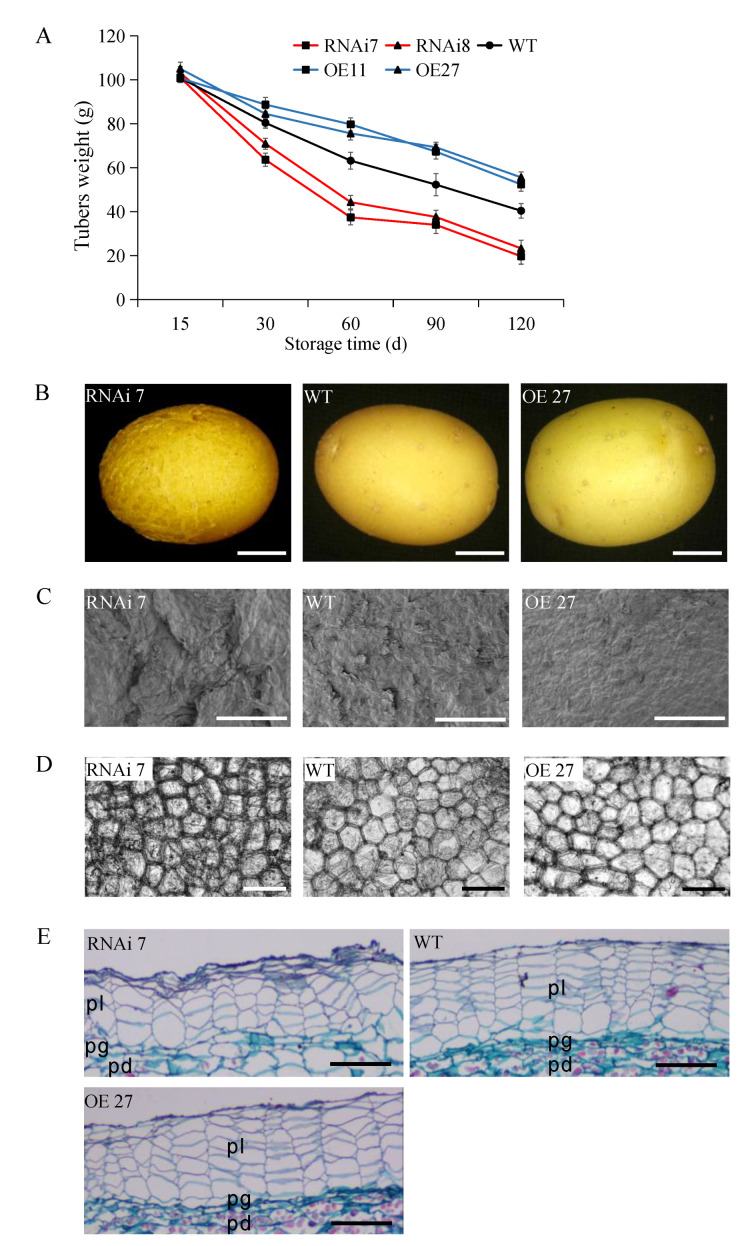
Morphological alterations in transgenic tubers. (**A**) Tuber weight of transgenic lines and wild-type (WT). (**B**) Tuber phenotype of RNAi line 7, OE line 27 and wild-type (WT) plants. Scale bar = 5 mm. (**C**) Scanning electron microscopy (SEM) micrograph of the skin surface from a wild-type and transgenic tuber. Scale bar = 500 μm. (**D**) Micrograph of different tuber skins. Scale bar = 100 μm. (**E**) Microscopy of cross sections of transgenic tuber and wild-type (WT). pl, phellem; pg, phellogen; pd, phelloderm. Scale bar = 100 μm.

**Figure 4 ijms-22-02287-f004:**
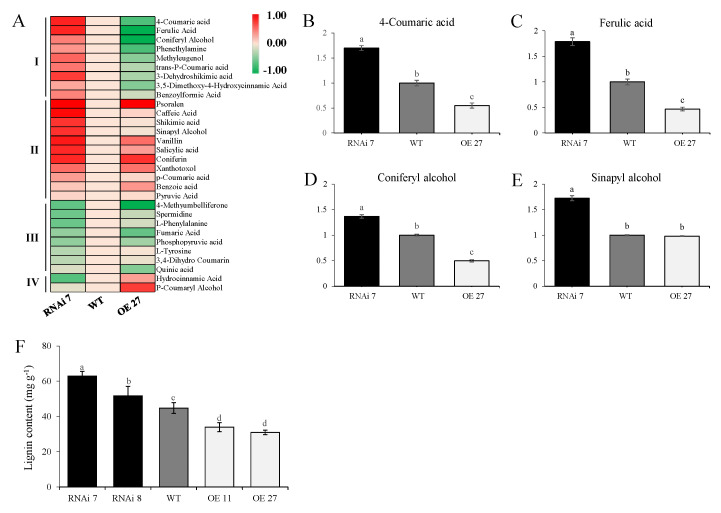
Lignin-related metabolites of control and *StSN2* transgenic tuber periderm. (**A**) Heat maps of differentially accumulated metabolites (DAMs) related to lignin synthesis. Relative contents of four key precursors, 4-Coumaric acid (**B**), Ferulic acid (**C**), Coniferyl alcohol (**D**) and Sinapyl alcohol (**E**), for lignin synthesis were identified and quantified (LC-MS/MS). (**F**) Lignin content in the RNAi line, OE line and wild-type (WT) periderm. The data are means ± SD of four biological experiments with triplicate measurements in each experiment. Different letters indicate a significant difference at *p* < 0.05 among wild-type, overexpression and RNAi potato plants.

**Figure 5 ijms-22-02287-f005:**
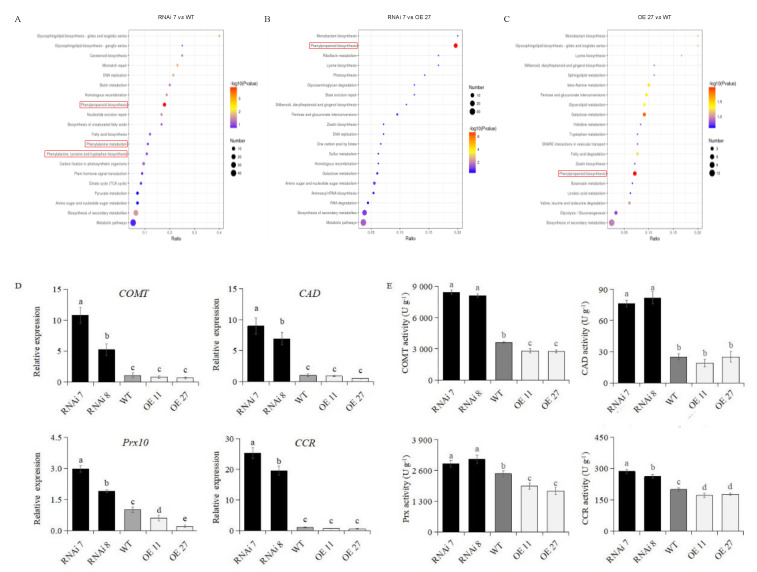
KEGG enrichment and analysis of lignin biosynthesis genes expression and enzyme activities. KEGG enrichment analyses of potato skins in three comparisons, RNAi 7 vs. WT (**A**), RNAi7 vs. OE27 (**B**) and OE27 vs. WT (**C**), at the 45th day of storage. The red rectangle highlights the lignin metabolism pathway. Gene expression (**D**) and enzyme activities (**E**) of COMT, CAD, Prx and CCR were investigated by qPCR assay and enzyme activities assay kits, respectively, at the 45th day. The data are means ± SD of four biological experiments with triplicate measurements in each experiment. Different letters indicate significant differences at *p* < 0.05.

**Figure 6 ijms-22-02287-f006:**
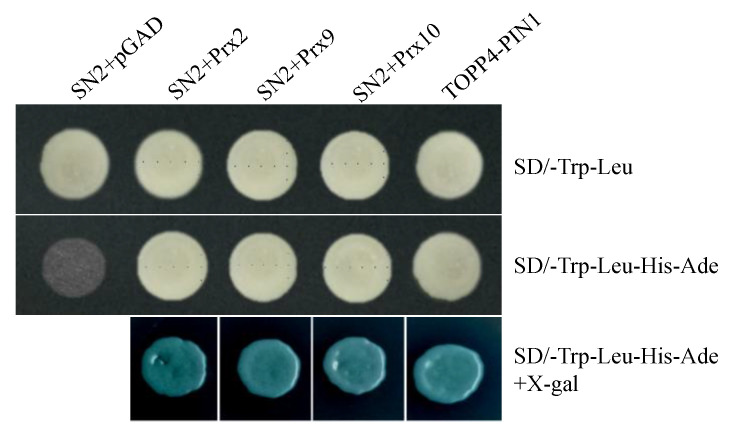
StSN2 physically interacts with peroxidases in a yeast two-hybrid system. The three peroxidases fused to pGADT7 are Prx2 (M1AU65), Prx9 (M1A251) and Prx10 (M1CCJ9). TOPP4, type-one protein phosphatase 4 in Arabidopsis; PIN1, pin-formed1 in Arabidopsis. X-gal, 5-Bromo-4-chloro-3-indolyl-b-D-galactopyranoside acid. The experiments were repeated three times.

**Figure 7 ijms-22-02287-f007:**
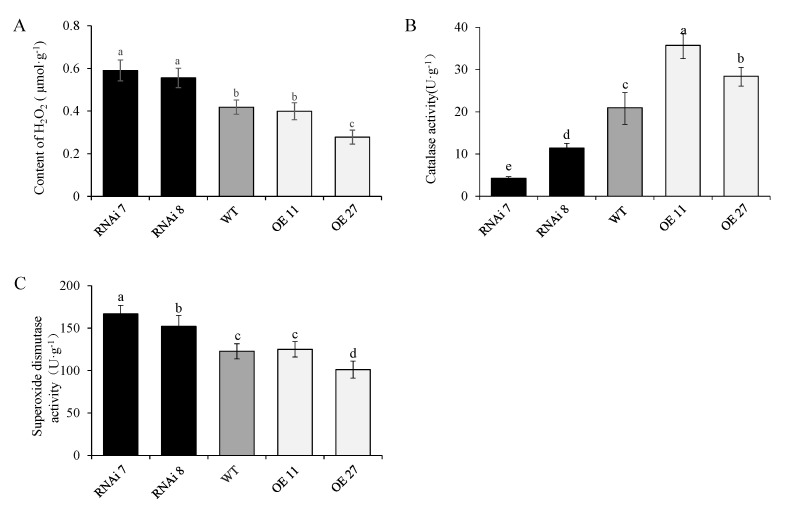
Effect of StSN2 on H_2_O_2_ content (**A**), superoxide dismutase (SOD) activity (**B**) and catalase (CAT) activity (**C**) in the three periderms. Data are means ± SD of three biological replicates. Different lowercase letters indicate a significant difference at *p* < 0.05 by Duncan’s test analysis.

## Data Availability

The data presented in this study are available on request from the corresponding author.

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
