# Peer review of "The Cysteine-Rich Peptide Snakin-2 Negatively Regulates Tubers Sprouting through Modulating Lignin Biosynthesis and H2O2 Accumulation in Potato"

_ijms, 2021, doi:10.3390/ijms22052287_

Round 1

Reviewer 1 Report

Comments to ijms-1091865

Previous studies by the authors have suggested that StSN2 may be associated with potato tuber sprouting and dormancy in transcriptome and proteome analyses. In this paper, they created over-expressed and down-regulated StSN2 mutant potatoes and observed their phenotypes. They also observed the formation of tuber periderm tissues and found relationship among lignin and dormancy state. Finally, they suggested that StSN2 affects lignin accumulation by interacting with a peroxidase as one of the molecular mechanisms of the relationship between StSN2 and dormancy/sprouting. Many reasonable experiments have been conducted, and I think the content would reach good levels for publication in the journal.

However, the text has many minor mistakes and text discrepancies that should be corrected before submission, and its qualities as a s academic manuscript are suspected. In addition to the points shown below, there may be other sentences that are not well-formatted in the text. So, the authors should read them carefully, correct them, and resubmit them.

The number of lines indicates the number of the PDF version (not the word file)

Line 117: Is this explanation from Figure 1B? If so, the figure legend and the number of days do not match.

Line 152: I don't know what the impurities are. It should be explained in the figure and in the figure legend.

Line 154: I don't know where the phellem cell layers are. Should be explained in the figure and in the figure legend.

Line 181: Figure 4B-E

Line 185: Figure 4F

Line 222: This sentence does not match the preceding and following sentences. Isn't it a mistake?

Line 384: The word “respectively” does not match the sentence.

Line 424: will be-> were

Figure 2B: It shows the transcription and protein accumulation of SN2, but controls should be shown together. It is impossible to make a quantitative judgment without control. The last two sentences of the Figure legend should come after (A).

Figure 4 and the text that describes it: It shows the lignin and its precursors in the SN2 mutants, but there is a lack of explanation as to why SN2 and the lignin pathway are involved. In addition, tissue observation of mutant tubers should be shown to see whether there is a difference in lignin accumulation.

Figure 5: In figure legend, the explanation of (D) in Figure legend is strange.

Table S1-S5 could not be seen (it was not included in the downloaded ijms-1091865-supplementary folder)

In the whole text, there are some sentences that do not start with capital letters.

Author Response

Dear Reviewer,

We had revised our manuscript in according to the comments, and upload the revised manuscript and supplementary folder. Please see the attachment. If you have any questions, please don't hesitate to contact us. Thank you!

Reviewer 2 Report

In this manuscript Wang and co-authors reveal and discuss the thorough investigation about the function and effect of the Snakin-2 (SN2) peptide on potato tuber dormancy and sprouting. Overall, the authors made a good and profound work on showing how the expression of SN2 manipulate the activity of relevant enzymes and consequently the biosynthesis of lignin. I find this manuscript suitable for publication in IJMS, after some minor corrections:

  • Though I downloaded the supplementary material, I could not find any Table S (Table S1-S5).
  • I have encountered a large number of linguistic errors, these should be corrected.
  • I have failed to understand the numberings such as "RNAi-SN2-#7" or "OE-SN2-#11". What do numberings following the hashtags stand for? I imagine some explanation may be found in the missing part of the supplementary information. 

Author Response

Dear Reviewer,

We had revised our manuscript in according to the comments, and up-loaded the revised manuscript and supplementary folder. Please see the attachment. If you have any questions, please don't hesitate to contact us. Thank you!

Round 2

Reviewer 1 Report

In the revised version, the parts I pointed out were corrected appropriately, and the other corrections also deepened the understanding of the contents of the manuscript. Therefore, I think that this manuscript would be acceptable.

There is small mistake, please check that.

--There is no figure legend for Figure S3 (E)

Author Response

Dear Reviewer,

We had corrected the mistake according to the suggestions, and the revisions were clearly highlighted in red  in figure legend for Figure S3 (E). If you have any questions, please don't hesitate to contact us. Thank you!

Kind regards

Xiyao Wang